# Mixed-Solvent Polarity-Assisted Phase Transition of Cesium Lead Halide Perovskite Nanocrystals with Improved Stability at Room Temperature

**DOI:** 10.3390/nano9111537

**Published:** 2019-10-30

**Authors:** Rui Yun, Li Luo, Jingqi He, Jiaxi Wang, Xiaofen Li, Weiren Zhao, Zhaogang Nie, Zhiping Lin

**Affiliations:** School of Physics and Optoelectronic Engineering, Guangdong University of Technology, Guangzhou 510006, China; 13570980324@163.com (R.Y.); 13690103732@163.com (J.H.); 18635208773@163.com (J.W.); selfiemua@163.com (X.L.); zwrab@163.com (W.Z.); zgniegdut@163.com (Z.N.); zhipinglphy@gdut.edu.cn (Z.L.)

**Keywords:** CsPbBr_3_, CsPb_2_Br_5_, solvent polarity, CTAB, phase transition

## Abstract

Cesium lead halide perovskite nanocrystals (NCs) have attracted enormous interest in light-emitting diode, photodetector and low-threshold lasing application in terms of their unique optical and electrical performance. However, little attention has been paid to other structures associated with CsPbBr_3_, such as CsPb_2_Br_5_. Herein, we realize a facile method to prepare dual-phase NCs with improved stability against polar solvents by replacing conventional oleylamine with cetyltrimethyl ammonium bromide (CTAB) in the reprecipitation process. The growth of NCs can be regulated with different ratios of toluene and ethanol depending on solvent polarity, which not only obtains NCs with different sizes and morphologies, but also controls phase transition between orthorhombic CsPbBr_3_ and tetragonal CsPb_2_Br_5_. The photoluminescence (PL) and defect density calculated exhibit considerable solvent polarity dependence, which is ascribed to solvent polarity affecting the ability of CTAB to passivate surface defects and improve stoichiometry in the system. This new synthetic method of perovskite material will be helpful for further studies in the field of lighting and detectors.

## 1. Introduction

Cesium lead halide perovskite (CsPbX_3_ X = Cl, Br, I) nanocrystals (NCs) have attracted enormous attention, having emerged as promising materials in the field of displays, lighting, lasing and photodetection [1,2]. A large number of studies on the thin-film, micro-structure and single crystals of these materials have been devoted to technological explorations for diverse applications [3,4], based on their outstanding optoelectronic performance, including ultrahigh photoluminescence quantum yield, narrow-band emission, flexible wavelength, high charge carrier mobilities and facile synthetic process [5,6]. In addition, other phases of cesium lead halide perovskite derivatives such as hexagonal Cs_4_PbBr_6_ and tetragonal CsPb_2_Br_5_ are observed in the form of quantum dots and so on [7,8], which possess structure-dependent physical properties and greatly expands the potential applications in sensing, catalysis, electro-chemistry and optoelectronics [9,10].

Recently, some reports indicated that CsPb_2_Br_5_ played an important role in improving the emission lifetime and stability of CsPbBr_3_ and enhancing solar cell efficiency [11]. Even so, studies on controllable syntheses of CsPb_2_Br_5_ and the mechanism of photoluminescence are still not abundant, especially in the area of theoretical simulation that cannot match the experimental results [12]. Some studies insisted on the pure CsPb_2_Br_5_ microplates with a bandgap of 2.44 eV exhibited lasing emission under both one- and two-photon excitation [13]. Jiang’s group indicated that the strong green emission is originated from coexisting phase CsPbBr_3_ rather than CsPb_2_Br_5_ and concluded that CsPb_2_Br_5_ is an indirect bandgap semiconductor with a bandgap of 3.1 eV and has high nonradiative Auger recombination, indicating that no luminescence will be generated from the CsPb_2_Br_5_ [12]. Some investigations have reported that the emissive CsPb_2_Br_5_ is associated to the sub-bandgap defects such as Br vacancy or Pb and Cs vacancies [14,15,16], while other researchers have proposed that the lead bromide complex in CsPb_2_Br_5_ is the reason for the luminescence [17,18]. Therefore, the research of the luminescence mechanism on CsPb_2_Br_5_ cannot be neglected.

Cesium lead halide perovskite NCs can rapidly nucleate and grow during synthesis, which is assigned to its low formation energy and fast crystallization rate [19]. What’s more, the crystalline phases are extremely sensitive to the ratios of the elements in the precursors, the post-processing and the film-formation [20]. Therefore, several strategies have been developed to control the composition, morphology and size of NCs in this rapid reaction by changing the experimental parameters. Jiang et al. found a phase transition from orthorhombic CsPbBr_3_ to tetragonal CsPb_2_Br_5_ and a shape evolution from octagonal to square by controlling the reaction time [12]. Deng et al. prepared uniform CsPb_2_Br_5_ nanowires and nanosheets with superior stability and high yield by mediating the ligands at room temperature [21]. Sun et al. synthesized dual-phase CsPbBr_3_-CsPb_2_Br_5_ composites at lower temperature and employed them as an emitting layer in LEDs, which exhibits a distinct improvement of about 21- and 18-fold in CE and EQE compared with reported CsPbBr_3_ LEDs [11]. However, little work has been devoted to the investigation of solvent polarity. The major role of the solvent is affecting the charge transfer rates between NCs and the surface bonding structure [22]. This is the first report where a systematic study that the morphology, phase structure and PL of NCs have a regular change under variable polar conditions. Such a study would trigger a deeper consideration of solvent effects and provide a new direction for improving optoelectronic device performance.

In this work, we demonstrate a new approach to synthesize NCs with cetyltrimethyl ammonium bromide (CTAB) instead of oleylamine under variable polar conditions. The role of CTAB in the system is discussed and NCs synthesized with it show enhanced stability against ethanol. Subsequently, we have revealed the phase structure of NCs transit from orthorhombic CsPbBr_3_ to tetragonal CsPb_2_Br_5_ with the solvent polarity increase and solvent polarity dependence of PL and defect density. Furthermore, the possible mechanism for NCs by combining ligand CTAB with mixed-solvent polarity is investigated. This would provide new guidance to modify the reprecipitation method.

## 2. Materials and Methods 

### 2.1. Materials

PbBr_2_ (Macklin, 99.0%, Shanghai, China), CsBr (Macklin, 99.5%), PbI_2_ (Macklin, 98.0%), CsI (Macklin, 99.9%), PbCl_2_ (Macklin, 99.5%), CsCl (Macklin, AR), cetyltrimethyl ammonium bromide (CTAB, Macklin, 99.0%), oleic acid (OA, Aladdin, AR, Shanghai, China), oleylamine (OAm, Aladdin, AR), dimethylformamide, (DMF, Aladdin, 99.8%), ethanol and toluene were purchased from Guangzhou Chemical Reagent Co. Ltd (Guangzhou, China) and were used directly without further purification.

### 2.2. Methods

#### 2.2.1. Synthesis of CsPbBr_3_/CsPb_2_Br_5_ NCs

In the typical synthesis of CsPbBr_3_/CsPb_2_Br_5_ QDs, PbBr_2_ (0.2 mmol) and CsBr (0.2 mmol) were dissolved in DMF (5 mL) at room temperature, OA (0.3 mL) and CTAB (0.05 g) were added to the constantly vigorous stirred DMF solution for 1h. Finally, 1 mL of mixed precursor solution was mixed in a new beaker with toluene (20 mL) under vigorous stirring for 20 min. The as-synthesized QDs were dispersed in toluene for further characterization.

#### 2.2.2. Synthesis of CsPbBr_3_/CsPb_2_Br_5_ NCs with Ethanol

Similar as the above CsPbBr_3_/CsPb_2_Br_5_ NCs approach, 1 mL of mixed precursor solution was injected into a new beaker with a mixture (20 mL) of ethanol (E) and toluene (T) (E:T = 0.2, 0.3, 0.4, 0.5, 0.6, 0.7) under vigorous stirring for 20 min.

#### 2.2.3. Synthesis of CsPbX_3_/CsPb_2_X_5_ NCs

Similar as the above CsPbBr_3_/CsPb_2_Br_5_ NCs approach, CsPb(Cl/Br)_3_/CsPb_2_(Cl/Br)_5_ NCs were prepared by PbCl_2_ (0.07 mmol), CsCl (0.07 mmol), PbBr_2_ (0.10 mmol), CsBr (0.10 mmol), OA (0.30 mL) and CTAB (0.02 g); CsPb(Br/I)_3_/CsPb_2_(Br/I)_5_ NCs were prepared by PbBr_2_ (0.10 mmol), CsBr (0.10 mmol), PbI_2_ (0.10 mmol), CsI (0.10 mmol), OA (0.30 mL) and CTAB (0.02 g).

#### 2.2.4. Synthesis of CsPbBr_3_/CsPb_2_Br_5_ NCs with OAm

Similar as the above CsPbBr_3_/CsPb_2_Br_5_ NCs approach, the CTAB were replaced with OAm (0.5 mL). 1 mL of mixed precursor solution was mixed in a new beaker with a mixture (20 mL) of ethanol (E) and toluene (T) (E:T = 0, 0.2, 0.3, 0.4, 0.5, 0.6, 0.7) under vigorous stirring for 20 min.

### 2.3. Characterization

Photoluminescence (PL) spectra were acquired on a fluorescence spectrophotometer (F-7000, Hitachi, Tokyo, Japan). Ultraviolet and visible absorption (UV-vis) spectra were measured with a UV-3600 plus spectrophotometer (Shimadzu, Kyoto, Japan). The absolute PLQYs were obtained on a integration sphere (mod. 2100, Otsuka Electronics, Tokyo, Japan). Fluorescence lifetimes were gained using a FM-4P time-corrected single-photon-counting (TCSPC) system (Horiba, Kyoto, Japan) at an excitation wavelength of 325 nm. FTIR spectra were measured on a Nicolet instrument (iS5, Madison, WI, United States) in the region of 3200–900 cm^−1^. X-ray diffractometry (XRD) patterns were collected with a D8 Advanced X-ray diffractometer (Bruker, Karlsruhe, Germany) using Cu Kα radiation (wavelength 1.55406 Å). Transmission electron microscopy (TEM) was performed on a JEM electron microscope (2100F, Tokyo, Japan) operating at 200 kV and energy dispersive spectra (EDS) were obtained with EDAX Genesis XM2 spectrometry. X-ray photoelectron spectroscopy (XPS) were recorded with an Escalab 250Xi X-ray photoelectron spectrometer (Thermo Fisher, Waltham, MA, United States) in the 3900–750 eV region.

## 3. Results

It is well known that surface ligands have a major impact on the shape, size and composition of NCs, and the size and morphology can be correlated with the performance of NCs in optics and electricity due to changes in band structure [23]. CTAB is one of the most common surfactants in the synthesis of gold nanorods, which is attributed to its electrostatic interaction with NCs [24,25]. Moreover, it’s generally accepted that the negative exciton trapping effect of Br vacancies (V_Br_) generated before nucleation cannot compensate for missing Br ions due to the fast nucleation rate, leading to a large amount of V_Br_ and some researchers have suggested the reduced V_Br_ density by passivation would lead to a higher QY [26,27]. The CTAB-modified NCs exhibit enhanced stability against polar solvents due to avoiding the ligand loss and low stability caused by the interligand proton transfer between oleylamine (OLA) and oleic acid (OA) [28]. Therefore, we explore a new synthetic method to trigger a deeper consideration of the nucleation and growth mechanism of NCs.

Furthermore, the effect of the solvent environment on the dispersibility, stability and photoelectric properties of the NCs has been widely investigated [29]. The strategy for the synthesis of CsPbBr_3_/CsPb_2_Br_5_ NCs are carried out by tuning solvent polarity, that is to change the faction of ethanol and toluene in the system. The main reason why ethanol was chosen as a solvent polarity regulator is that CTAB has a higher solubility in ethanol, leading to the concentration of Br only slightly changing [30]. The crystallized phases with different morphology are obtained by mixing a precursor in good solvent (N,N-dimethylformamide, DMF) into a poor mixture under ambient conditions at room temperature. The details for synthesis can be found in the Methods section and Appendix A.

Figure 1 shows the transmission electron microscopy (TEM) images and high-resolution TEM (HRTEM) images of the representative samples a and d, which were synthesized with a poor mixture at V_E_: V_T_ = 0 and 0.4, respectively. It is observed that the NCs of 8–22 nm are uniform and monodispersed in sample a (Figure 1a,b), while Figure 1c,d show spherical nanoparticles (NPs) of 3–4 nm uniformly embedded in NCs with a similar size distribution (Appendix A) and the yellow circles indicate the position of the embedded NPs. With an increase of the solvent polarity, a significant difference in shape is presented in Appendix A, from which we can see that the NCs around 13.5 nm are sharply reduced to be spherical quantum dots of 1-4 nm after adding ethanol (Figure 1b). The size of the NCs grows dramatically to about 24.3 nm (Appendix A) and then reduce slightly to about 14.7 nm (Appendix A). Subsequently, the embedded NPs vanish and the dispersion of NCs becomes worse after slight adjustment of the ratio between E and T (Appendix A). The length of NCs continues to increase and reaches a maximum of approximately 120 nm (Appendix A). Finally, agglomerated NCs of about 30 nm in size are obtained when V_E_:V_T_ = 0.7. (Appendix A). This reveals that the solvent polarity has an important role in the final morphology of products [31] and implies changes in the structural phase. It is worth noting that the overall size of NCs shows a gradual increase during the process and the dispersion becomes worse as the solvent polarity increases, which could be related to the decrease of ligand efficiency caused by excessive ethanol [32].

The HRTEM images reveal the CsPb_2_Br_5_ structure with the lattice fringe spacing of 0.3 nm and 0.42 nm as shown in Appendix A. Furthermore, the selected area electron diffraction (SAED) patterns of sample a, where the lattice fringe spacings of 0.59 nm, 0.24 nm and 0.21 nm correspond to the (2,2,2), (1,3,2) and (4,0,0) crystal planes of CsPbBr_3_ and the lattice fringe spacing of 0.43 nm and 0.30 nm are associated with the (2,0,0) and (2,2,0) crystal planes of CsPb_2_Br_5_, respectively. These results suggest that CsPbBr_3_ and CsPb_2_Br_5_ phases coexist in these regions. What is more, the NPs (Appendix A) with the lattice fringe spacing of 0.228 nm assigned to CsPbBr_3_ and the NCs corresponds to CsPb_2_Br_5_. It is confirmed that the CsPbBr_3_ NPs are uniformly embedded in the CsPb_2_Br_5_ NCs.

In order to investigate the effect of solvent on the composition of NCs, XRD measurements for the above seven samples were performed as shown in Figure 2, where the diffraction patterns from NCs are matched well with the main diffraction peaks at 15.21, 21.64, 30.70° corresponding to (110), (200), (220) plane of the orthorhombic CsPbBr_3_ (JCPDS No. 01-072-7929) in the yellow area and 11.67, 23.39, 35.44, 37.90, 47.86° corresponding to (002), (210), (312), (313) and (420) plane of the tetragonal CsPb_2_Br_5_ (JCPDS No. 00-025-0211) in the blue area, respectively. It is easy to see that the peaks in the blue region, especially at 15.21° and 30.70°, are gradually weakened until V_E_:V_T_ = 0.6 and then almost disappear. However, the sharp and intense peaks in 11.67° are obviously enhanced when V_E_:V_T_ > 0.2. Furthermore, the percentage of CsPbBr_3_ and CsPb_2_Br_5_ in a mixture can be roughly estimated by the ratio of their strongest XRD peaks. It’s obvious that the percentage of CsPbBr_3_ decreased from about 66% to 7% corresponding to samples a-g, while the content of CsPb_2_Br_5_ increased from around 34% to nearly pure phase. The above observations demonstrate that the solvent polarity controls the molar ratio of CsPbBr_3_/CsPb_2_Br_5_ in the composite NCs and ethanol has a positive effect on phase transition.

The absorption spectra (Figure 3a) were measured to gain more insight into the degree of electronic disorder in the crystals, which is attributed to the fact that the absorption edge is known as the Urbach tail. The Urbach energy (*E_U_*) reflects the cumulative effect of impurities, defects and electron-phonon interactions on NCs, which could be obtained by fitting the Urbach tails in the logarithmic absorption spectra according to the Urbach’s rule [33]:(1)$$\alpha \left(E\right){=\alpha }_{0}\exp\left[\frac{\left({E-E}_{0}\right)}{E_{U}}\right]$$

We obtain *E_U_* = k_B_*T*/*σ*(*T*), where k_B_ is the Boltzmann constant, *T* is the absolute temperature and *σ* is the steepness parameter [34]. Figure 3b displays the *E_U_* value of NCs synthesized in a mixture, which was obtained by fitting curves. It’s observed that the *E_U_* gradually reduces to a minimum (9.54 meV) and the maximum value reached is approximately 26.38 meV, which reveals the solvent polarity dependence upon the *E_U_*. The NCs with lower *E_U_* means that they possess a lower degree of structurual disorder and/or defect density than other NCs [35]. This indicates that the solvent polarity is a key factor to control internal defect density or structural disorder of NCs during the reprecipitation process. Furthermore, some investigations proved that the Br^—^ concentration in the octahedron can be characterized by the red-shift of the absorption spectra [36]. Therefore, we can conclude that solvent polarity may be useful to control the CTAB passivation effect on NCs.

Figure 3c shows photoluminescence (PL) spectra of the CsPbBr_3_/CsPb_2_Br_5_ NCs synthesized in different mixtures with a strong green emission, which varies slightly in PL intensity, central wavelength and full width at half-maximum (FWHM). The specific relationship between V_E_/V_T_ and three parameters are shown in Figure 3d. It can be seen that the PL intensity increases first and reaches its maximum when the ratio between CsPb_2_Br_5_ and CsPbBr_3_ phase is around 4.6 (Figure 2), and then the decrease of PL intensity when this ratio excess 4.6. The significant improvement in PL intensity of samples c-e is associated with the suitable volume fraction ratio of both structures [37]. Appendix A shows the NCs synthesized with oleic acid and oleylamine quench after adding ethanol, which demonstrates that the NCs synthesized with CTAB have solvent-resistant ability as described in previous reports. Another solvent effect on NCs is the Stokes’ shift of 8 nm, and the small Stokes’ shift originates from band edge radiative recombination [38]. It’s worth noting that the FWHM of the emission peak in sample a-g is between 21 nm to 29 nm, which roughly agrees with the narrow size distribution of the NCs (Appendix A).

The PL decays and lifetime obtained by triexponential decay functions are shown in Figure 3e. The triexponential functions (Appendix A) and specific data obtained are recorded in Appendix A. It’s observed that the sample d has a longer lifetime (18.20 ns) than sample a (9.68 ns). Some reports have mentioned that the lifetime is decreased with the increase in hydrogen solvent polarity and inferred that solvent polarity plays an important role in changing the NCs trap states [39].

To further investigate the composition and phase transitions process, the film formed by samples a and d on the glass were characterized by X-ray photoelectron spectroscopy (XPS). All XPS spectra were calibrated with C 1s peak at 284.6 eV. Figure 4 shows the XPS survey spectra and high-resolution XPS spectra of sample a and d at Cs, Pb, Br. It can be seen that the peaks of Cs 3d, Pb 4f and Br 3d all are shifted to lower binding energy (BE) after adding ethanol. Pb^2+^ into NCs is in two chemical environments. The BE curves of Pb 4f_5/2_ and Pb 4f_7/2_ located at approximate 143 eV and 138 eV. It is noteworthy that the peaks marked as pink and orange after fitting are ascribed to the surface Pb ions and their areas occupied are smaller compared to sample a, implying the V_Br_ defects in sample d being reduced under higher polar condition [40]. Similarly, Br in NCs also exists in two chemical environments, and the BE curves of Br 3d_3/2_ and Br 3d_5/2_ appearing at approximately 68.5 eV and 67 eV are assigned to Pb-Br and Cs-Br. The significant differences of phases with different proportions are ascribed to their bond and structure [41]. Furthermore, the element ratio of Cs to Pb obtained by XPS (Appendix A) is about 1:1 (sample a) and 1:2 (sample d), which is in good agreement with EDS results (Appendix A), while the excessive Br is originated from CTAB.

Moreover, the Fourier transform infrared (FTIR) spectra show the surface groups of NCs synthesized in different polar condition, as shown in Figure 5a. The peaks located at 2980 cm^−1^ and 2895 cm^−1^ are due to ν(C-H) in the -CH_2_ group [29]. The intense peaks at 1679, 1394 cm^−1^ and 1265 cm^−1^ are assigned to ν(C=O). It’s worth noting that the C=O bond is obviously shifted to a lower frequency with the increase of the solvent polarity, as shown in Figure 5b, which could be attributed to one dimensional structures formed by the coordination between one DMF molecule and Pb [42]. The peaks appearing at 1095 cm^−1^ and 1053 cm^−1^ are originated from C-N stretching vibrations in CTAB molecules [43]. Figure 5b shows highly magnified FTIR spectra in the 1000–1200 cm^−1^ region, in which the 1053 cm^−1^ peak of sample a isn’t obvious, while the 1053 cm^−1^ peak exists in samples with ethanol. These four samples have similar peaks positions and no significant change in transmittance value. It is shown that the change in PL intensity is mainly associated with its defect density, rather than the charge transfer rates between NCs and the surface bonds.

The sharp emission peak of NCs can be tuned from 458 nm to 600 nm by changing the halogen ratio and UV/Vis spectra exhibit intense absorption, as shown in Figure 6a. Besides, the PL spectra of CsPbBr_3_/CsPb_2_Br_5_ NCs synthesized with ethanol, isopropyl alcohol, cyclohexane, hexane, ether, ethyl acetate, methanol, acetone and toluene in a ratio of 0.4 were measured and are presented in Figure 6b. The PL of initially NCs is quenched after the addition of polar or non-polar solvent, which could be attributed to several reasons: the introduction of some functional groups causes a decrease in carrier mobility [44] and even the crystal structure is destroyed due to the nature of the ionic lattice and highly dynamic ligands process [45]. 

Sehrawat et al. indicated that the variation in PL properties could be demonstrated by geminate recombination and an associated variation in Onsager length related to the dielectric constant [46]. However, a significant improvement in the PL of NCs formed in toluene and ethanol (Appendix A) is due to CTAB dissolved in ethanol will ionize into CTA^+^ and Br^—^ [30] and the higher Br^—^ concentration in the system could improve the internal defect of NCs and stoichiometry. Hyun et al. proposed that solvent molecules can affect the charge transfer process by intervening the dielectric layer or the rearrangement of solvent molecules on the surface of NCs [47]. Majima et al. revealed that a strong solvent-polarity dependence on the electron-transfer process [48]. Therefore, it can be inferred that the solvent effects on NCs can’t be neglected from the variable performance of NCs synthesized under different polar conditions.

## 4. Conclusions

In conclusion, the solvent polarity-assisted transition from dual-phase to CsPb_2_Br_5_ phase offers a technique to alter the morphology of NCs. Such a phase transition could be related to two reasons: the degree of CTAB dissolution and growth of NCs under different polar conditions. The obtained NCs show enhanced stability and solvent polarity dependence of PL intensity, which could be assigned to the fact that CTAB molecules are highly soluble in ethanol and the produced Br^—^ can effectively passivate defects and improve the stoichiometry in the system. These guesses can be proved by defect density calculated in absorption spectra. Therefore, we can conclude that solvent polarity affects the ability of CTAB to passivate surface defects and it’s a key factor for the final performance of the resulting NCs. This work provides new insights for deeper understanding in the field of perovskite NCs.

## Figures and Tables

**Figure 1 nanomaterials-09-01537-f001:**
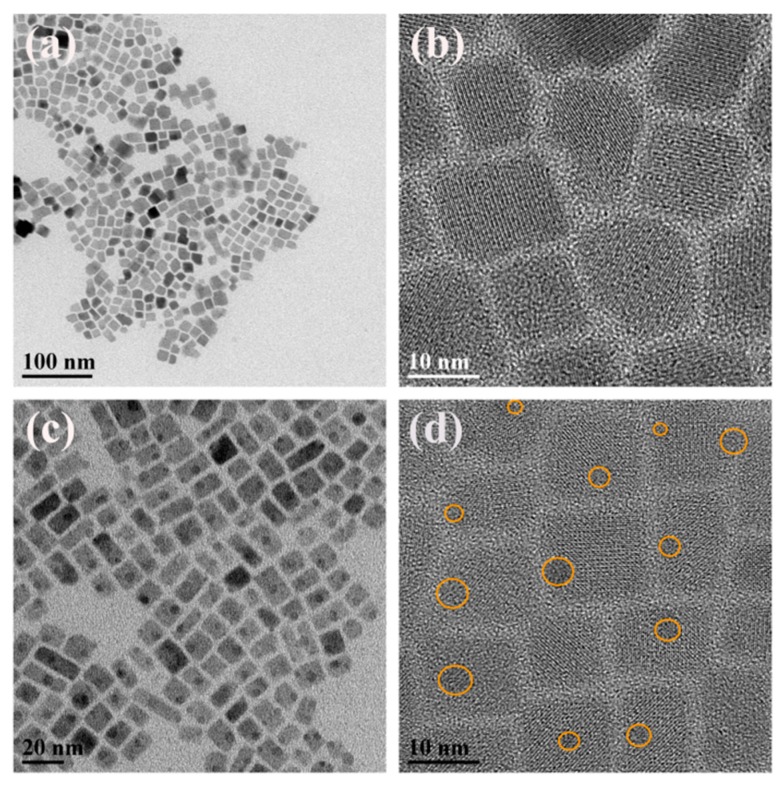
(**a**) Typical TEM overview image of sample a; (**b**) HRTEM image of selected lager sample a; (**c**) Typical TEM overview image of sample d; (**d**) HRTEM image of selected lager sample d.

**Figure 2 nanomaterials-09-01537-f002:**
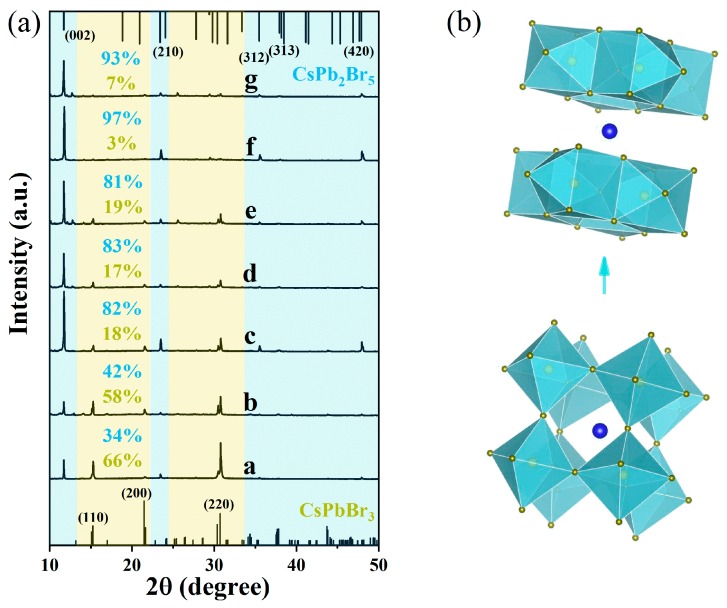
(**a**) X-ray diffraction (XRD) patterns of sample a-g; (**b**) Schematic representation of mixed-solvent polarity assisted the transition of orthorhombic CsPbBr_3_ to tetragonal CsPb_2_Br_5_.

**Figure 3 nanomaterials-09-01537-f003:**
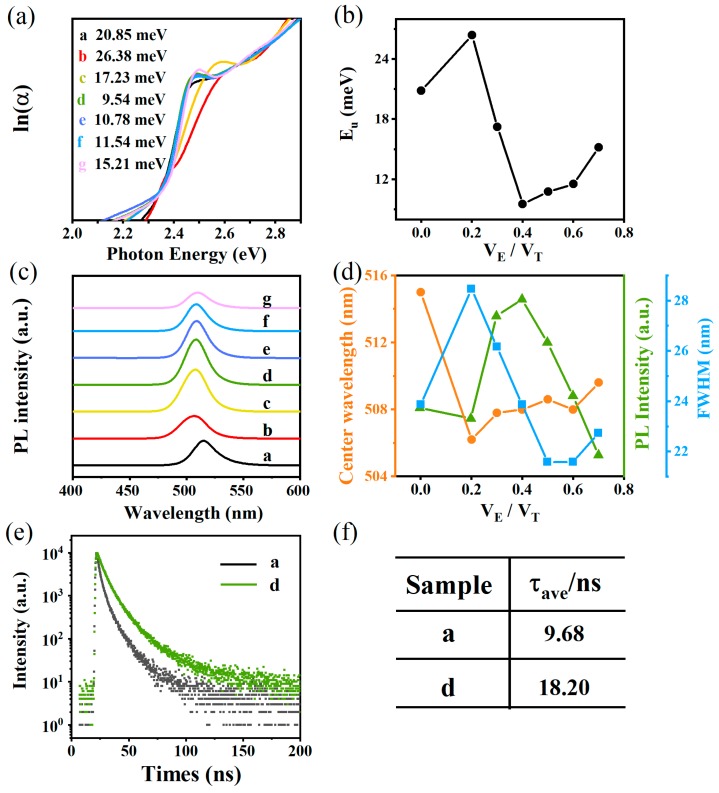
(**a**) Logarithmic absorption coefficient of sample a-g as a function of photon energy; (**b**) Relationship between VE/VT and Urbach energy; (**c**) Emission spectra of samples a-g excited by 365 nm; (**d**) Relationship between VE/VT and Central wavelength, PL intensity and FWHM; (**e**) PL decay for samples a and d with a 325 nm pulse laser; (**f**) The fitted average lifetime of samples a and d.

**Figure 4 nanomaterials-09-01537-f004:**
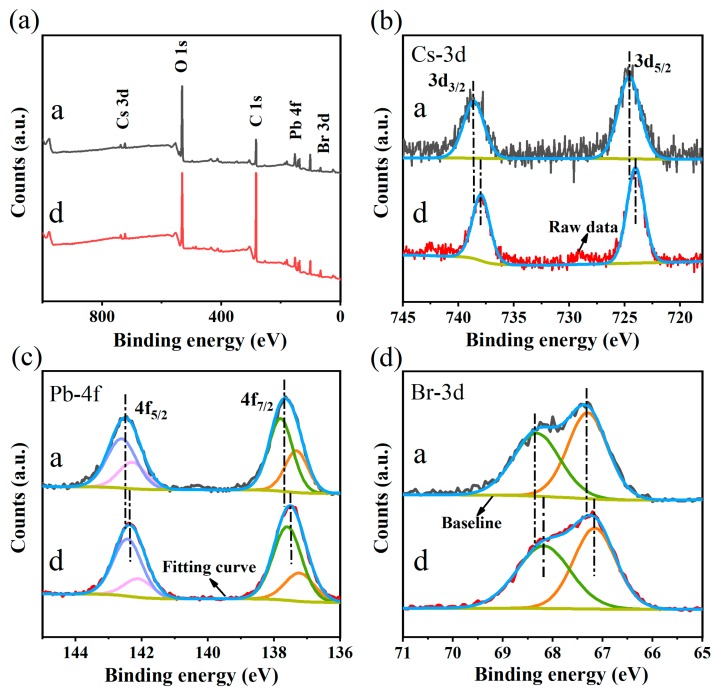
(**a**) XPS survey spectra of sample a film; (**b**) high-resolution XPS spectra of sample a and d at Cs; (**c**) high-resolution XPS spectra of sample a and d at Pb; (**d**) high-resolution XPS spectra of sample a and d at Br.

**Figure 5 nanomaterials-09-01537-f005:**
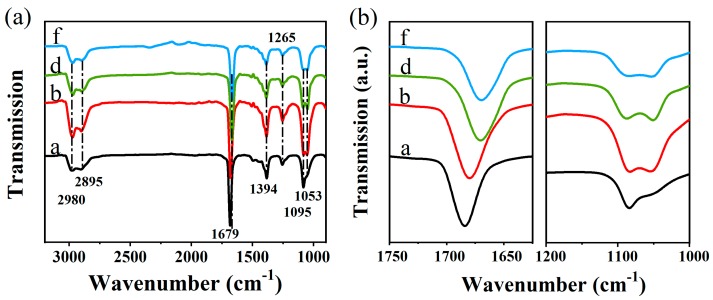
(**a**) Fourier transform infrared (FTIR) spectra of samples a, b, d, f; (**b**) Highly magnified FTIR spectra of samples a, b, d, f ranging from 1750 to 1625 cm^−1^ and 1200 to 1000 cm^−1^.

**Figure 6 nanomaterials-09-01537-f006:**
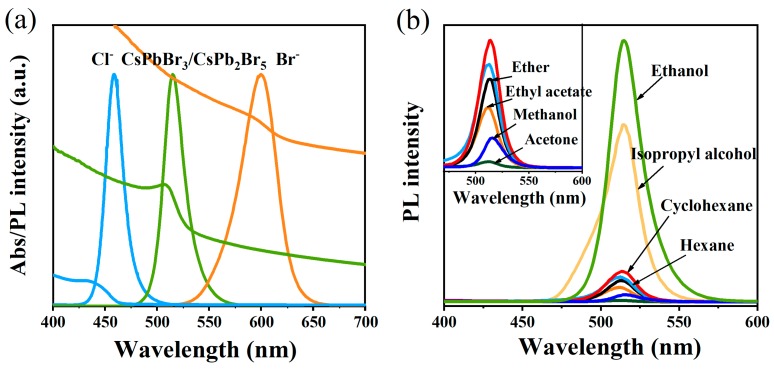
(**a**) UV-vis absorption and PL spectra of NCs synthesized with pure toluene; (**b**) PL spectra of NCs synthesized with different organic solvents: toluene = 0.4.

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
