# Peer review of "Mixed-Solvent Polarity-Assisted Phase Transition of Cesium Lead Halide Perovskite Nanocrystals with Improved Stability at Room Temperature"

_nanomaterials, 2019, doi:10.3390/nano9111537_

Round 1

Reviewer 1 Report

Herein, the authors propose a new, facile, synthetic method to prepare dual-phase perovskite nanocrystals with enhanced stability against polar solvents by replacing oleylamine with CTAB in the reprecipitation process. Regulation of the NC growth was observed upon changes in the mixed solvent polarity employed, manifested with different NC sizes, morphologies and controling the phase transition between orthorombic and tetragonal NC phases. A strong solvent polarity dependence of the PL spectra and the NC defect density was obtained as solvent polarity affects the ability of CTAB to passivate NC surface defects.  

The method proposed is new and, overall, the work is nicely organized and clearly presented. The obtained results are clearly important for the field of perovskite NCs as effective passivation of surface defects is a major obstacle for their further development. There is just one comment I have regarding Fig. 6b which relates to the possibility of extracting a relation between the  PL spectral peak and the solvent dielectric constant.

Furthermore, I'd like to note that the manuscript requires editing by the authors in order to improve the English used and correct some mistakes (for example, NC particle instead of partical) prior to publication.

Author Response

Response: Thank you very much for your positive comments. We have added and rephrased as "The PL of initially NCs is quenched after the addition of polar or non-polar solvent, which could be attributed to several reasons: the introduction of some functional groups causes a decrease in carrier mobility and even the crystal structure is destroyed due to the nature of the ionic lattice and highly dynamic ligands process. In addition, Sehrawat et al indicated that the variation in PL properties could be demonstrated by geminate recombination and an associated variation in Onsager length related to the dielectric constant" according to the referee’s comment. See also text in lines 258-264 of page 9 in the revised manuscript. Meanwhile, we have made changes in Figure S1, S2, S3 and S5 according to the referee’s suggestion and checked carefully in the whole paper.

Reviewer 2 Report

The reviewed paper deal with synthesis, structural and optical properties of cesium lead halide perovskites. The paper show interesting results, but before publication several corrections should be applied:

In figure caption (Fig.1.) authors should explain meaning of the yellow circles. As authors don’t change te volume of oleic acid and CTAB in the table S1 it’s enough to show only the ethanol/toluene volume ratio Authors should calculate ratio of CsPbBr3 to CsPb2Br5 phase using the XRD patterns. It would help to analyze also luminescence spectra that are related to the measured structure. In the analysis of the emission spectra it can be seen the non-regularity. This may be associated to the volume fraction ratio of both structures. Please make such as analysis. Reference 33 not define equation for calculation of the Urbach energy. In this reference only some assumptions were made and therefore authors should refer to another source. In the equation is not present Eu (Urbach energy) and other energies (E, E0) appearing in the eq. are not described. VE/VT is not a good parameter that is used for analysis of the luminescence spectra. In my opinion here should be used parameter CsPbBr3/CsPb2Br5 phase volume ratio, or particle size of the NC. S5 “Particle size” instead of “partical size” Why decays were measured only for two samples? It’s difficult to define some relation between luminescence spectra and decay times (as well as for VE/VT ratio) Authors suggest that “higher Br- concentration in the system could improve the internal defect of NCs.”. I would suggest that higher Br- lead to improving stoichiometry in the system and higher crystallization of the expected structure.

If the authors correct above mentioned issues the manuscript may be published in the Nanomaterials.

Author Response

Reply to the Comments of Referee #2 

Comment #1

The reviewed paper deal with the synthesis, structural and optical properties of cesium lead halide perovskites. The paper shows interesting results, but before publication several corrections should be applied.

Response: We greatly appreciate the referee for his/her positive comments on our manuscript and the efforts to improve the quality of our manuscript. 

Comment #2

If the authors correct above mentioned issues the manuscript may be published in the Nanomaterials. 

Response: Thank you very much for your positive comments. We have made all the requested changes that have been reflected either in the text. Some incorrect discussion or inappropriate expression has been deleted or rephrased throughout the manuscript. The following is the point-to-point response to the comments listed in the report. 

Comment #3

In figure caption (Fig.1.) authors should explain meaning of the yellow circles.  

Response: Sorry for our not mentioned description of the yellow circles. we have added and rephrased as "and the yellow circles indicate the position of the embedded NPs." See also text in lines 139-140 of page 4 in the revised manuscript.

Comment #4

As authors don’t change te volume of oleic acid and CTAB in the table S1 it’s enough to show only the ethanol/toluene volume ratio. 

Response: Thank you for reminding us of the unnecessary writing about the ligands part. To compare the synthesis condition of NCs, we have deleted the content of oleic acid and CTAB in the table S1 for clear expression. 

Comment #5

Authors should calculate ratio of CsPbBr3 to CsPb2Br5 phase using the XRD patterns. It would help to analyze also luminescence spectra that are related to the measured structure. In the analysis of the emission spectra it can be seen the non-regularity. This may be associated to the volume fraction ratio of both structures. Please make such as analysis.

Response: Thank you for this valuable suggestion. We agree that it may be more appropriate to express luminescence spectra with the volume fraction ratio of both structures. The percentage of CsPbBr3 and CsPb2Br5 in a mixture were estimated and the concrete contents were marked in Figure 2a. Meanwhile, we have added and rephrased as "Furthermore, the percentage of CsPbBr3 and CsPb2Br5 in a mixture is roughly estimated by the ratio of their strongest XRD peak. It,s obvious that the percentage of CsPbBr3 decreased from about 66% to 7% corresponding to sample a-g, while the content of CsPb2Br5 increased from around 34% to nearly pure phase". See also text in lines 172-175 of page 5 in the revised manuscript. More analysis on the relation between the volume fraction ratio and luminescence spectra can be observed in Comment #7. 

Comment #6

Reference 33 not define equation for calculation of the Urbach energy. In this reference only some assumptions were made and therefore authors should refer to another source. In the equation is not present Eu (Urbach energy) and other energies (E, E0) appearing in the eq. are not described.

Response: Thank you for your suggestion. We have replaced the original reference with a new, which defines an equation for calculation of the Urbach energy. The equation is changed as (see also text in lines 186 of page 6) and the calculation of Urbach energy: EU=kBT/σ(T) are extracted from the above equation. Therefore, we didn’t explain too much for other energies (E, E0) appearing in the equation. 

Comment #7

VE/VT is not a good parameter that is used for analysis of the luminescence spectra. In my opinion here should be used parameter CsPbBr3/CsPb2Br5 phase volume ratio, or particle size of the NC.

Response: Many thanks for pointing out this important issue. We agree that it may be more appropriate to analyze luminescence spectra with the CsPbBr3/CsPb2Br5 phase volume ratio. We have added and rephrased as "It can be seen that the PL intensity increases first and reaches the maximum when the ratio between CsPb2Br5 and CsPbBr3 phase is around 4.6 (Figure 2), and then the decrease of PL intensity when this ratio excess 4.6. The significant improvement in PL intensity of sample c-e is associated with the suitable volume fraction ratio of both structures. Figure S4 shows the NCs synthesized with oleic acid and oleylamine quenches after adding ethanol" according to the referee’s comment. See also text in lines 199-203 of page 7 in the revised manuscript.

Comment #8

S5 “Particle size” instead of “partical size”.

Response: Thank you for reminding us of this writing. We have checked carefully in the whole paper and made changes in Figure S1, S2, S3 and S5 according to the referee’s suggestion.

Comment #9

Why decays were measured only for two samples?

Response: Thank you for your suggestion. The origin experiment is mainly centered at samples a and d, but we found that it’ s rather difficult to analyze and eliminate contingency. So the further experimental results were obtained by narrowing the variable gap. However, because of the instrument failure, it is rather difficult for us to perform further characterization of all samples.

Comment #10

It’s difficult to define some relation between luminescence spectra and decay times (as well as for VE/VT ratio). 

Response: Sorry for our not serious description. We realize that the above description might be confusing and misleading to readers as indicated by the referee. Based on the simple characterization, It’s rather difficult to define some relation between luminescence spectra and decay times (as well as for VE/VT ratio). Therefore, we have deleted the inappropriate text description and rephrased as "The PL decays and lifetime obtained by triexponential decay functions are shown in Figure 3e. The triexponential functions (equ S1, S2) and specific data obtained are recorded in Table S2. It’s observed that the sample d has a longer lifetime (18.20 ns) than sample a (9.68 ns). Some reports mentioned that the lifetime is decreased with the increase in hydrogen solvent polarity and inferred that solvent polarity plays an important role in changing the NCs trap states". in lines 209-213 of page 7 in the manuscript. 

Comment #11

Authors suggest that “higher Br- concentration in the system could improve the internal defect of NCs.”. I would suggest that higher Br- lead to improving stoichiometry in the system and higher crystallization of the expected structure. 

Response: Many thanks for pointing out this important issue. The defects play a significant role for particles with nanoscale dimensions due to the high surface to volume ratio. Many investigates have shown VBr is the most abundant defect in NCs due to its lower formation energy compared with VCs and VPb[1]. Moreover, it’s general accepted that the negative exciton trapping effect of VBr generated before nucleation cannot compensate the missing of Br ions due to the fast nucleation rate, leading to a large amount of VBr and some researches suggested the reduced VBr density by passivation would lead to a higher QY[2,3]. Therefore, we thought “higher Br- concentration in the system could improve the internal defect of NCs”. In addition, the NCs halide-deficient could be caused by a stoichiometric amount of the halide salt when PbX2 is employed as the only halogen source, leading to leaving the excessive Pb atoms. The Pb atoms would be the media of nonradiative recombination, which has passive effects on the PL properties of NCs[4]. Hence, we approve that “higher Br- lead to improving stoichiometry in the system and achieving a more perfect crystal structure”. We have added the details according to the referee’s comment, see also text in lines 22 of page 1, lines 266 of page 9 and lines 282 of page 10 in the revised manuscript. 

Reference

[1] Kang, J.; Wang, L.W. High defect tolerance in lead halide perovskite CsPbBr3. J. Phys. Chem. Lett. 2017, 8, 489-493

[2] Wu, Y.; Wei, C.; Li, X.; Li, Y.; Qiu, S.; Shen, W.; Cai, B.; Sun, Z.; Yang, D.; Deng, Z.; Zeng, H. In situ passivation of PbBr64- octahedra toward blue luminescent CsPbBr3 nanoplatelets with near 100% absolute quantum yield. ACS. Energy. Lett. 2018, 3, 2030-2037.

[3] Pan, J.; Quan, L.N.; Zhao, Y.; Peng, W.; Murali, B.; Sarmah, S.P.; Yuan, M.; Sinatra, L.; Alyami, N.M.; Liu, J.; Yassitepe, E.; Yang, Z.; Voznyy, O.; Comin, R.; Hedhili, M.N.; Mohammed, O.F.; Lu, Z.H.; Kim, D.H.; Sargent, E.H.; Bakr, O.M. Highly efficient perovskite-quantum-dot light-emitting diodes by surface engineering. Adv. Mater. 2016, 28, 8718–8725

[4] Ahmed, T.; Seth, S.; Samanta, A. Boosting the photoluminescence of CsPbX3 (X = Cl, Br, I) perovskite nanocrystals covering a wide wavelength range by postsynthetic treatment with tetrafluoroborate salts. Chem. Mater. 2018, 30, 3633-3637.

Reviewer 3 Report

In the manuscript "Mixed-Solvent Polarity Assisted Phase Transition of Cesium Lead Halide Perovskite Nanocrystals with Improved Stability at Room Temperature" the authors propose to carry out the synthesis of perovskite nanocrystals using CTAB as a ligand and mixture of good/bad solvents - toluene/ethanol. They show that by varying the ratio between the solvents one can achieve nanocrystals of different phases, CsPbBr3 and CsPb2Br5 having different optical properties probably due to the altered ligands ability to passivate the surface. The study uses various classical characterisation methods to show the solvent effects on chemical, structural, and optical properties of the nanocrystals. Even though the effect of solvent polarity on the synthesis of the NCs is probably present, the consequences and importance of this finding are limited - the modification of the NCs does not allow to open important new perspectives nor it present a significant scientific novelty, which could be useful for the scientific community. A series of similar studies have been previously published and nowadays the mechanism of the NC formation and surface passivation is relatively well established. In addition, the manuscript style is below average, several phrases and discussion are hard to understand, which complicates the appreciation of the article.

Overall, I do not believe that the manuscript provides important scientific novelty and thus it should not be published in Nanomaterials.

Author Response

Response: Thank you for this suggestion. the effect of solvent polarity exists in the synthesis of NCs as you mentioned, and the crystalline phases are extremely sensitive to the ratios of the elements in the precursors, the post-processing and the film-formation[1]. Although solvent environment is not prominent compared with the several conventional factors (ligands, concentration, reaction temperature, time) in the studies on perovskite NCs, but it is a factor that cannot be ignored in the synthesis of NCs as shown in manuscript and solvent environment plays an important role in the synthesis of other NCs. Zheng et al demonstrated that amphiphilic carbon dots synthesized in different solvent could be tuned from blue to orange emission, which is sensitive and promising for the detection of volatile organic compounds[2]; Jiang et al introduced a method for preparing quantum dot thin film using a mixed-solvent system to achieve better light-emitting devices[3]; Kirmani et al proposed that understanding the effect of solvent environment on colloidal-quantum-dot solar-cell can help improve the overall performance of the device[4]. Therefore, we hope to introduce our findings from this perspective and provide a new direction to study the variation of NCs in many aspects.

In our paper, as under-coordinated lead atoms (VBr) are the most abundant active defects in NCs due to its low formation energy, new strategies for synthesis employing halide-rich reaction conditions have been developed[5]. We attempt to replace the common ligands with CTAB, which achieves a enhanced stability owing to the fact that avoiding the interligand proton transfer between oleylamine and oleic acid[6]. In order to explore the effect of solvent on NCs, the NCs synthesized in various polar conditions exhibit a strong considerable solvent polarity dependence of size, morphology, structure and luminescence property, the calculated defect density show a trend by changing the solvent environment corresponding to the variable volume fraction ratio of CsPbBr3/CsPb2Br5 and PL property. Moreover, many researchers have mentioned that the better NCs are prepared with a certain percentage of the “good” solvent/“poor” solvent/“clean” solvent by adjusting ligands/concentration/ reaction temperature/time, but few people have really explained why NCs are synthesized with this ratio of reagents. Therefore, we focus on the effect of solvent environment on the growth and properties of NCs, and obtain enhanced stability and strong luminescence. It’s meaning and valuable for deeper understanding in the field of perovskite NCs. In order to improve the quality of the article, we made further changes to the content and grammar of the article.

Thank you again for your comments!

Reference

[1] Palazon, F.; Urso, C.;Trizio, L.D.; Akkerman, Q.; Marras, S.; Locardi, F.; Nelli, I.; Ferretti, M.; Prato, M.; Manna, L. Postsynthesis transformation of insulating Cs4PbBr6 nanocrystals into bright perovskite CsPbBr3 through physical and chemical extraction of CsBr. Acs. Energy. Lett. 2017, 2, 2445-2448.

[2] Zheng, M.; Li, Y.; Zhang, Y.; Xie, h. Solvatochromic fluorescent carbon dots as optic noses for sensing volatile organic compounds. Rsc. Adv. 2016, 6, 83501–83504.

[3] Jiang, C.; Zhong, Z.; Liu, B.; He, Z.; Zou, J.; Wang, L.; Wang, J.; Peng, J.; Cao, Y. Coffee-ring-free quantum dot thin film using inkjet printing from a mixed-solvent system on modified ZnO transport layer for lightemitting devices. Acs. Appl. Mater. Inter. 2016, 8, 26162-26168.

[4] Kirmani, A.R.; Carey, G.H.; Abdelsamie, M.; Yan, B.; Cha, D.; Rollny, L.R.; Cui, X.; Sargent, E.H. Amassian, A. Effect of solvent environment on colloidal-quantum-dot solar-cell manufacturability and performance. Adv. Mater. 2014, 26, 4717-4723.

[5] Seth, S.; Ahmed, T.; De, A.; Samanta, A. Tackling the defects, stability and photoluminescence of CsPbX3 perovskite nanocrystals. Acs. Energy. Lett. 2019, 47, 1610-1618.

[6] Cai, Y.; Wang, L.; Zhou, T.; Zheng, P.; Li, Y.; Xie, R.J. Improved stability of CsPbBr3 perovskite quantum dots achieved by suppressing interligand proton transfer and applying a polystyrene coating. Nanoscale. 2018, 10, 21441–21450.

Round 2

Reviewer 2 Report

The paper may be published in the present form